# Using Random Effects to Account for High-Cardinality Categorical Features and Repeated Measures in Deep Neural Networks

**Giora Simchoni**
Department of Statistics
Tel Aviv University
Tel Aviv, Israel, 69978
gsimchoni@tauex.tau.ac.il

**Saharon Rosset**
Department of Statistics
Tel Aviv University
Tel Aviv, Israel, 69978
saharon@tauex.tau.ac.il

## Abstract

High-cardinality categorical features are a major challenge for machine learning methods in general and for deep learning in particular. Existing solutions such as one-hot encoding and entity embeddings can be hard to scale when the cardinality is very high, require much space, are hard to interpret or may overfit the data. A special scenario of interest is that of repeated measures, where the categorical feature is the identity of the individual or object, and each object is measured several times, possibly under different conditions (values of the other features). We propose accounting for high-cardinality categorical features as random effects variables in a regression setting, and consequently adopt the corresponding negative log likelihood loss from the linear mixed models (LMM) statistical literature and integrate it in a deep learning framework. We test our model which we call LMMNN on simulated as well as real datasets with a single categorical feature with high cardinality, using various baseline neural networks architectures such as convolutional networks and LSTM, and various applications in e-commerce, healthcare and computer vision. Our results show that treating high-cardinality categorical features as random effects leads to a significant improvement in prediction performance compared to state of the art alternatives. Potential extensions such as accounting for multiple categorical features and classification settings are discussed. Our code and simulations are available at `https://github.com/gsimchoni/lmmnn`

## 1 Introduction

In recent years deep neural networks (DNNs) have served as the method of choice for learning complex non-linear relations between features of large datasets and for hard prediction tasks. Yet, despite their advanced machinery such as stochastic gradient descent (SGD) and convolutional layers, DNNs (as other machine learning tools) do not readily lend themselves to handling categorical features of high cardinality. Such features are often seen in modeling tasks, a good recent example would be Lin et al. [16] who used electronic medical records (EMR) of hospital patients to predict hospital readmission, and had to deal with the patient's diagnosis as a categorical feature with thousands of levels. In a recent review Hancock and Khoshgoftaar [13] surveyed the approaches taken by DNN practitioners for turning the levels of categorical features into numeric form, of which the most used approaches are one-hot encoding (OHE) and entity embeddings. OHE and embeddings also appear as the go-to solution in one of the most cited handbooks of machine learning by Géron [9]. See Section 4 for a more detailed discussion of these approaches.

A different approach for handling high-cardinality categorical features is that of linear mixed models (LMM), see e.g. McCulloch et al. [18]. In the LMM framework statisticians differentiate between

*fixed effects* (FE) which are regular parameters to be estimated from the data and *random effects* (RE), which are parameters that are not directly estimated, but rather treated as random variables, and their distribution is estimated instead. This mechanism is often used to deal with high cardinality categorical variables or "cluster" indicators. In a typical setting a statistician would be interested in the effect few *designed* experimental conditions have on a dependent variable, e.g. the effect of treatment A versus treatment B on a patient's blood pressure. The treatment effect would be considered a fixed effect, while the effect of the city in which a subject resides — a categorical feature with possibly thousands of levels — would be considered a RE variable, often having normal distribution $\mathbb{N}\left(0, \sigma_b^2\right)$ with $\sigma_b^2$ being an unknown "variance component" to be estimated (see Section 2 for a formal description). Another common example of a RE variable is the subject of the experiment herself, in a *repeated measures* setting, where a subject is treated to both treatments A and B. In such a setting a statistician would usually deem a subject's effect as random, i.e. coming from a possibly infinite population of subjects, and use proper LMM machinery to handle such a variable.

In this paper we are interested in examining modern big-data predictive modeling domains, where DNNs would be a natural choice of modeling approach, but where high dimensional categorical variables or repeated measures are an important aspect of prediction. Examples we pursue might include: predicting the gaze direction of a person's eyes from repeated facial images of the same individuals [24], detection of diabetic retinopathy from retinal fundus photographs where each patient is photographed for both left and right eyes [11], prediction of a person's fluid intelligence from dozens of features including a person's job which can take hundreds of levels [14] and predicting the price of an Airbnb rental with host being one of tens of thousands [15].

To address such challenges we seek a simple extension to DNN which allows to treat high-cardinality categorical features as RE and ultimately improve the network's prediction performance in a regression setting. We do this by generalizing the LMM-framework negative log likelihood (NLL) function and treating it as a natural loss function for the entire network, minimizing both the "fixed" and "random" parts, and improving the bottom line test mean squared error (MSE). We demonstrate the superiority of our method over using existing methods such as OHE, entity embeddings and standard LMM via R's lme4 package (Bates et al. [2]), as well as previous attempts at integrating RE in DNN (MeNets, Xiong et al. [22]), on both simulated and real datasets from various applications such as healthcare and computer vision. Finally we discuss future directions, such as using RE in DNN in a classification setting and with more complex covariance structures such as kriging.

## 2 A Brief Tour of Linear Mixed Models

The canonical linear model assumes:
$$y = X\beta + \epsilon, \tag{1}$$
where $X$ is the $n \times p$ model matrix where the $i$th row is $x_i$, $\beta \in \mathbb{R}^p$ is a vector of model parameters to be estimated and $\epsilon \in \mathbb{R}^n$ is normal i.i.d noise, i.e. $\epsilon \sim \mathbb{N}\left(0, \sigma_e^2 I\right)$, where $\sigma_e^2$ is a variance parameter and $I$ is the $n \times n$ identity matrix.

A linear *mixed* model treats the $\beta$ parameters as *fixed* effects, and allows additional data to enter the model in the form of a $n \times q$ matrix $Z$ and a vector of *random* effects $b \in \mathbb{R}^q$:
$$y = X\beta + Zb + \epsilon \tag{2}$$
Here $b$ are random variables, typically assumed to have a multivariate normal distribution $\mathbb{N}\left(0, D\right)$ where $D$ is a $q \times q$ positive semi-definite matrix holding usually unknown *variance components* to be estimated, let these be $\theta$, so $D$ could be written as $D(\theta)$. The structure of this covariance matrix is up to the researcher but there are typically simplified structures used. It is further assumed that there is no dependence between the normal noise and the random effects, i.e. $cov\left(\epsilon, b\right) = 0$.

Let us write the vector of all variance components as $\psi = [\sigma_e^2, \theta]$. How to estimate $\beta, \psi$? The statistical approach of choice is maximum likelihood estimation (MLE). We write the marginal distribution of $y$ as $\mathbb{N}\left(X\beta, V\right)$ where $V(\psi) = ZD(\theta)Z' + \sigma_e^2 I$, and from here it is straightforward to see that the log likelihood is:
$$l(\beta, \psi|y) = -\frac{1}{2}\left(y - X\beta\right)' V(\psi)^{-1}\left(y - X\beta\right) - \frac{1}{2}\log |V(\psi)| - \frac{n}{2}\log 2\pi \tag{3}$$
The MLEs for $\beta, \psi$ maximize $l(\beta, \psi|y)$ or equivalently minimize the *negative* log likelihood (NLL) in (3). Typically maximizing the likelihood is difficult with given constraints on $\psi$ (which are non-negative variance components), and to get less biased estimates for $\psi$ alternatives methods such as

restricted maximum likelihood estimation (REML) are preferred. For full details see e.g. McCulloch et al. [18].

Of particular interest is the *random intercepts* model, in which a single categorical random variable, such as "subject" or "city", is assumed. If this so-called "clustering" variable has $q$ levels, then $Z$ is an indicator matrix holding at entry $[i, j]$ 1 if observation $i$ belongs to cluster $j$ and zero otherwise. In this case it is often assumed that the random effects for this variable are independent of one another, hence: $D = \sigma_b^2 I$. In this setting $V(\psi) = \sigma_b^2 Z Z' + \sigma_e^2 I$ is a block diagonal matrix and its inversion can be avoided in (3). More on that in Section 3.

How to predict $\hat{y}_{te}$ in a machine learning scenario, where $(X, Z, y)$ are typically split into training and testing sets $(X_{tr}, Z_{tr}, y_{tr})$ and $(X_{te}, X_{te}, y_{te})$? In general one would use $y$'s conditional distribution:

$$\hat{y}_{te} = X_{te}\hat{\beta} + Z_{te}\hat{b}, \tag{4}$$

where $\hat{\beta} = (X_{tr}'\hat{V}^{-1}X_{tr})^{-1}X_{tr}'\hat{V}^{-1}y_{tr}$ are the estimated fixed effects once the estimated variance components $\hat{\psi}$ are input into $\hat{V}$, and:

$$\hat{b} = \hat{D}Z_{tr}'\hat{V}^{-1}\left(y_{tr} - X_{tr}\hat{\beta}\right) \tag{5}$$

is the so called *best linear unbiased predictor* (BLUP), as $b$ are not actually parameters to be estimated, but random variables to be predicted.

Sometimes, however, (4) is not possible such as in the case of the random intercepts model, where $Z_{te}$ holds levels of the random variable unseen before, e.g. a new subject. In this case it is customary to use $y$'s marginal distribution and predict $\hat{y}_{te}$ to be $X_{te}\hat{\beta}$, i.e. without the random part. Finally, still under the random intercepts model, it can be shown that for a given level $j$ and corresponding random effect $\hat{b}_j$, the computation can be simplified to avoid the inversion of $\hat{V}$ giving:

$$\hat{b}_j = \frac{n_j\hat{\sigma}_b^2}{\hat{\sigma}_e^2 + n_j\hat{\sigma}_b^2}\left(\bar{y}_{tr;j} - \overline{X_{tr}\beta}_j\right), \tag{6}$$

where $\left(\hat{\sigma}_e^2, \hat{\sigma}_b^2\right)$ are the estimated variance components, $n_j$ is the no. of observations in cluster $j$ and $\bar{y}_{tr;j}$ and $\overline{X_{tr}\beta}_j$ are the true and predicted mean values of $y$ in cluster $j$ respectively.

## 3  LMMNN: Our Proposed Approach

Let us change (2) into:

$$y = f(X) + g(Z)b + \epsilon, \tag{7}$$

where $f$ and $g$ are non-trivial functions which DNNs are suitable for, e.g. non-linear and involving interactions. We propose to use NLL as a natural loss function, where $X\beta$ is replaced by the DNN outputs $f(X)$:

$$NLL(f, g, \psi|y) = \frac{1}{2}(y - f(X))'V(g, \psi)^{-1}(y - f(X)) + \frac{1}{2}\log|V(g, \psi)| + \frac{n}{2}\log 2\pi, \tag{8}$$

where $V(g, \psi) = g(Z)D(\theta)g(Z)' + \sigma_e^2 I$. Note that $f$ and $g$ are kept general as possible, to allow any acceptable DNN architecture, including convolutional and recurrent neural networks. See Figure 1 for a schematic description of our approach which we call LMMNN, in the case $f$ and $g$ are approximated with a simple multilayer perceptron.

We propose using existing DNN machinery, mainly SGD, to fit the model and its variance components. To be more specific, we implemented a custom *NLL* loss layer where at each epoch (8) is calculated on a small batch typically of size 30-50 observations and auto-differentiation is handled by Keras [4]. An alternative could be using explicit formulas for the variance components $\psi$ derivatives in case $V(\psi)$ is simple:

$$\frac{\partial NLL}{\partial \psi} = -\frac{1}{2}(y - f(X))'V^{-1}\frac{\partial V}{\partial \psi}V^{-1}(y - f(X)) + \frac{1}{2}tr\left(V^{-1}\frac{\partial V}{\partial \psi}\right), \tag{9}$$

where $\frac{\partial V}{\partial \psi}$ is $g(Z)\frac{\partial D(\theta)}{\partial \theta}g(Z)'$ for the $\theta$ variance components and $I$ for $\sigma_e^2$.

The main challenge in implementing mini-batch SGD on (8) is the inverse and determinant of $V(g, \psi)$ which may not decompose into sums on batches. An important exception is the main setting of interest in our paper in the random intercepts model, when $g(Z) = Z$ and $V(g, \psi)$ is a block-diagonal matrix. If all blocks are identical, such that each level of the categorical feature has $m$ observations $(X_j, y_j)$ where $j = 1, \ldots q$, we can write $V(\psi) = diag(V_1, \ldots, V_q)$ where each $V_j$ block is of size $m \times m$ and $V(\psi)_j = \sigma_b^2 J_m + \sigma_e^2 I_m$ where $J_m$ is a $m \times m$ all 1s matrix. This means we can write the inverse in (8) as block diagonal as well, $V(\psi)^{-1} = diag(V_1^{-1}, \ldots, V_q^{-1})$, and the log determinant in (8) as a sum of log determinants: $\log |V(\psi)| = \sum_{j=1}^{q} \log |V_j|$. The NLL in (8) can now be written as a sum: $NLL(f, \psi | y) = \sum_{j=1}^{q} \frac{1}{2} (y_j - f(X_j))' V_j^{-1} (y_j - f(X_j)) + \frac{1}{2} \log |V_j| + \frac{m}{2} \log 2\pi$. Most importantly, the gradient in (9) can be decomposed into a sum of gradients over mini-batches of size $m$:

$$\frac{\partial NLL}{\partial \psi} = \sum_{j=1}^{q} \left[ -\frac{1}{2} (y_j - f(X_j))' V_j^{-1} \frac{\partial V_j}{\partial \psi} V_j^{-1} (y_j - f(X_j)) + \frac{1}{2} tr \left( V_j^{-1} \frac{\partial V_j}{\partial \psi} \right) \right] \quad (10)$$

If all blocks are not identical but small (the typical case), we can still make sure that mini-batches consist of each level $j$'s $n_j$ observations. For more general cases the mini-batch approach (where inverse and determinant are calculated on the mini-batch) still seems to work well in practice, some related theory can be found in Chen et al. [3], see Section 6 for further discussion on more LMM scenarios.

For prediction, we accommodate (4) as well:

$$\hat{y}_{te} = \hat{f}(X_{te}) + \hat{g}(Z_{te}) \hat{b}, \quad (11)$$

where $\hat{f}$ and $\hat{g}$ are the outputs of the DNNs used to approximate $f$ and $g$, and $\hat{b}$ are the predicted random effects with the modified BLUP: $\hat{b} = D(\hat{\psi}) \hat{g}(Z_{tr})' V(\hat{g}, \hat{\psi})^{-1} \left( y_{tr} - \hat{f}(X_{tr}) \right)$

Note further that in the likely case where $g$ is simply the identity matrix and we are interested in a single clustering variable as in the random intercepts model, we can also avoid inversion of $V$ in the loss function as described in [18], as well as in computing the modified BLUP:

$$\hat{b}_j = \frac{n_j \hat{\sigma}_b^2}{\hat{\sigma}_e^2 + n_j \hat{\sigma}_b^2} \left( \bar{y}_{tr;j} - \overline{\hat{f}(X_{tr})}_j \right) \quad (12)$$

## 4  Related Work

### 4.1  One-Hot Encoding and and Entity Embeddings

As Hancock and Khoshgoftaar [13] state, OHE is often a "first step", and the standard method for handling categorical features. If $v$ of length $n$ is a categorical feature of $q$ distinct levels, OHE would add $q$ binary features $z_1, \ldots, z_q$, one for each level, with $z_{li} = 1$ if observation $i$ has level $l$ in feature $v$, and 0 otherwise. The main advantages of the OHE approach are that it is deterministic, fast and highly explainable. On the downside it is hard to scale — as $q$ can be in the tens or hundreds of thousands as in the examples below — a burden to any algorithm even when accounting for sparse representation of features. A second disadvantage of OHE features is how little information each feature carries. Other methods for converting categorical features into numeric can reach a much more meaningful representation.

One such method is entity embeddings (See e.g. Guo and Berkhahn [12]). Entity embeddings start by feeding a neural network with feature $v$ after it has been one-hot encoded into a $q$-length vector, and using the network's inherent loss function and back propagation algorithm to learn a low-dimensional representation of length $d$ (typically $d << q$) to each of $v$'s $q$ levels. This results in a lookup table $E$ of dimensions $q \times d$ which serves the predictive modeling task and can also later be re-used via transfer learning where the representation learned for one task can serve for other tasks, see e.g. Do and Ng [6]. It is also our experience that the entity embeddings approach performs well. Apart from that, efficient implementations make sure it scales well when $q$ is large, unlike OHE the resulting

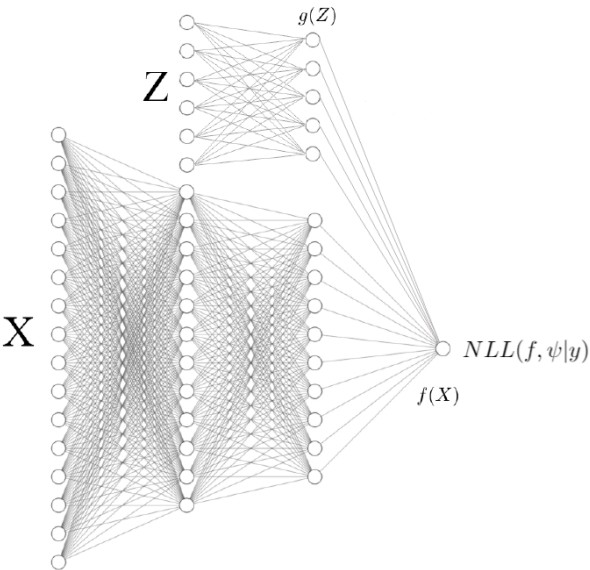

Figure 1: Schematic description of LMMNN using a simple deep MLP for fitting $f$ and $g$, and combining outputs with the NLL loss layer.

representations may carry a lot of information, and the option to perform transfer learning on these is appealing. On the other hand, entity embeddings may consume much space (the $E$ lookup table), they may need to be learned for each new task and the resulting representations are usually hard to interpret.

### 4.2 Previous Attempts at Combining Random Effects in DNN

There have been few previous attempts at combining random effects in DNN. The efforts we are aware of are MeNets by Xiong et al. [22] and DeepGLMM by Tran et al. [21].

**MeNets** Inspired by a non-linear mixed effects model $y = \nu\left(X\beta + Zb\right) + \epsilon$, where $\nu$ is some non-linear function, Xiong et al. [22] and Xiong et al. [23] propose the following model to learn fixed effects $\beta$ and random effects $b$:

$$y = f\left(X\right)\beta + f\left(X\right)b + \epsilon \tag{13}$$

There are two main differences between (13) and our (7): First, the $Z$ RE features matrix is missing, rather the RE features are *learned* as the output of the neural network to a standard input $X$, i.e $f(X)$. Second, (13) uses a *single* neural network to learn both the fixed and random features, where (7) is more general and allows different functions $f, g$ for fixed and random features respectively. In particular, (7) allows for $g$ to be the identity function which may certainly be appropriate for high cardinality categorical features and repeated measures and gives intuitive meaning to $b$.

In order to learn $\beta$ and $b$ the authors use variational expectation maximization (V-EM) combined with SGD: A E-Step in which $\hat{\beta}, \hat{b}$ are updated while minimizing the standard squared loss with a DNN, followed by a M-Step where the variance components $\hat{\psi}$ are updated so as to maximize a NLL loss similar to (8). Hence, an additional critique of MeNets could be that they essentially use two loss functions for a single task, whereas we use NLL as a single loss function to be maximized in the SGD framework, making our implementation simpler and easier to track. In addition, in order to update $\hat{b}_j$ at each E-step, there is the need to invert $\hat{V}_j$ which is the $j$th RE covariance matrix of size $n_j \times n_j$. This can be hard to scale if one of the categorical feature levels has a large $n_j$ [1]

---

[1]See Section 4 and the UKB dataset, when one of the levels of the categorical feature "job" has over 29,000 observations - being unemployed - and MeNets inverts a $29,000 \times 29,000$ matrix at each epoch.

Finally we note that MeNets were demonstrated in a limited context: the authors put special emphasis on computer vision applications, mainly that of gaze estimation, as a consequence the DNN architecture used by the authors is solely a convolutional network, whereas we demonstrate our results on a broad class of DNN architectures. Moreover, the categorical clustering feature MeNets used was the subject whose gaze was recorded, with typically only 10-50 subjects (and many repeated measures for each subject). Below we demonstrate MeNets in a wider context and compare it to LMMNN, illustrating the superiority of our approach in terms of both predictive performance and computation.

**DeepGLMM** A different approach taken by Tran et al. [21] is based on a very specific mixed effects model, in which each subject $i$ is repeatedly measured at different times $t$ for some response $y_{it}$ which can be continuous as well as discrete, as modeled by generalized linear models (GLM). In such a model it makes sense to not only have a random intercept for each subject but also a random slope $a_i$. The authors write:

$$g(\mu_{it}) = f(x_{it}^{(1)}, w, \beta^{(1)}) + (\beta^{(2)} + a_i)' x_{it}^{(2)}, \tag{14}$$

where $w$ are the network parameters, $x^{(1)}$ and $x^{(2)}$ are the features expected to have nonlinear and linear effects, respectively, $\beta^{(1)}$ and $\beta^{(2)}$ are the fixed nonlinear and linear effects respectively, and $\mu = E(y|x)$, via some link function $g$, e.g. the logit function for binary $y$. A Bayesian approach based on variational approximation is used to maximize the likelihood for (14).

We note that (14) is similar to our criterion in (7), when $g$ is the identity function and $y$ is linear in $Z$, the RE features matrix. We further acknowledge that our approach is only currently applicable for a continuous $y$ (though see future directions in Section 6). However, the variational approximation algorithm proposed in DeepGLMM, which combines numerous elements such as importance sampling, factor covariance, variable selection as well as choice of priors makes it challenging to implement, let alone use as a "plug-in" for different DNN architectures as we strive to do. Due to the high conceptual and computational complexity of DeepGLMM we were not able to implement and test it below like MeNets — it cannot scale to problems of the size we consider and to diverse architectures.

## 5 Results

### 5.1 Simulated Data

We start with a simple random intercepts model simulation, in which $n = 100,000$ and there is a single categorical random variable with $q$ levels ($q \in \{100, 1000, 10000\}$), such that $Z$ is a $n \times q$ binary matrix, and $b$ is a $q$-length vector of i.i.d random effects, sampled from a $\mathbb{N}\left(0, \sigma_b^2\right)$ distribution ($\sigma_b^2 \in \{0.1, 1, 10\}$). The $q$ levels are not evenly distributed among the $N$ observations, rather we use a multinomial distribution sampling, for more details see our code and description in additional material. There are 10 fixed features in $X$ non-linearly related to $y$:

$$y = (X_1 + \cdots + X_{10}) \cdot \cos(X_1 + \cdots + X_{10}) + 2 \cdot X_1 \cdot X_2 + Zb + \epsilon \tag{15}$$

Notice in this case $g$ is assumed to be the identity function. We sample $X$ features from a $\mathbb{U}(-1, 1)$ distribution, $\epsilon$ from a $\mathbb{N}\left(0, \sigma_e^2\right)$ distribution where we keep $\sigma_e^2 = 1$ always. We perform 5 iterations for each $(q, \sigma_b^2)$ combination (9 combinations in total), in which we sample the data, randomly split it into training (80%) and testing (20%), train our models to predict $\hat{y}_{te}$ and compare the bottom-line MSEs in predicting $y_{te}$. We compare our model's MSE to those of R's *lme4* package results (i.e. standard LMM), MeNets, OHE, entity embeddings and ignoring the categorical feature in $Z$ altogether. We use the same DNN architecture for all neural networks, that is 4 hidden layers with 100, 50, 25, 12 neurons, a ReLU activation and a Dropout of 25% in each, and a final output layer with a single neuron. The loss we use is squared error loss (MSE) for OHE, embeddings and ignoring the RE, and NLL for LMMNN and MeNets (as mentioned above, MeNets uses squared loss for estimating fixed effects and NLL for variance components only). For both losses we use a batch size of 30 and an early stopping rule where training is stopped if no improvement in validation loss is seen within 10 epochs. We initialize both $\hat{\sigma}_e^2, \hat{\sigma}_b^2$ to be 1.0 where appropriate: R's lme4 and LMMNN, and compare the resulting final estimates for these two methods. All runs are made on a Nvidia Quadro P620 GPU on a Windows machine, with Keras [4] and Tensorflow [1], with our own NLL loss layer for LMMNN, available on Github.

Table 1: Simulated model with $g(Z) = Z$, mean test MSEs and standard errors in parentheses. Bold results are non-inferior to the best result in a paired t-test. Hence, LMMNN is significantly better than all competitors in all scenarios.

| $\sigma_b^2$ | $q$ | Ignore | OHE | Embeddings | lme4 | MeNets | LMMNN |
|---|---|---|---|---|---|---|---|
| 0.1 | $10^2$ | 1.25 (.012) | 1.20 (.010) | 1.18 (.006) | 2.92 (.017) | 1.15 (.013) | **1.14 (.010)** |
| | $10^3$ | 1.23 (.009) | 1.31 (.008) | 1.24 (.004) | 2.96 (.022) | 1.40 (.065) | **1.14 (.009)** |
| | $10^4$ | 1.22 (.004) | 1.54 (.008) | 1.56 (.007) | 2.97 (.014) | 1.51 (.133) | **1.17 (.010)** |
| 1 | $10^2$ | 2.17 (.041) | 1.23 (.008) | 1.21 (.010) | 2.93 (.013) | 1.22 (.022) | **1.09 (.010)** |
| | $10^3$ | 2.16 (.015) | 1.39 (.015) | 1.32 (.014) | 2.94 (.013) | 1.42 (.091) | **1.14 (.006)** |
| | $10^4$ | 2.14 (.013) | 1.68 (.013) | 1.68 (.013) | 3.17 (.021) | 1.66 (.056) | **1.27 (.014)** |
| 10 | $10^2$ | 10.45 (.38) | 1.56 (.044) | 1.57 (.039) | 2.92 (.012) | 1.86 (.156) | **1.10 (.013)** |
| | $10^3$ | 11.37 (.11) | 1.75 (.022) | 1.72 (.041) | 2.95 (.024) | 2.19 (.143) | **1.12 (.015)** |
| | $10^4$ | 11.31 (.04) | 2.34 (.027) | 2.20 (.033) | 3.37 (.020) | 3.29 (.423) | **1.32 (.007)** |

Table 2: Simulated model, estimated variance components on average

| | | $g(Z) = Z$ | | | | $g(Z) = ZW$ | | | |
|---|---|---|---|---|---|---|---|---|---|
| | | lme4 | | LMMNN | | lme4 | | LMMNN | |
| $\sigma_b^2$ | $q$ | $\hat{\sigma}_e^2$ | $\hat{\sigma}_b^2$ | $\hat{\sigma}_e^2$ | $\hat{\sigma}_b^2$ | $\hat{\sigma}_e^2$ | $\hat{\sigma}_b^2$ | $\hat{\sigma}_e^2$ | $\hat{\sigma}_b^2$ |
| 0.1 | $10^2$ | 2.92 | 0.09 | 1.12 | 0.10 | 2.92 | 0.49 | 1.09 | 0.15 |
| | $10^3$ | 2.91 | 0.10 | 1.16 | 0.10 | 2.91 | 3.52 | 0.92 | 0.16 |
| | $10^4$ | 2.90 | 0.10 | 1.17 | 0.17 | 2.91 | 33.8 | 0.19 | 0.16 |
| 1 | $10^2$ | 2.90 | 1.01 | 1.12 | 1.00 | 2.91 | 2.44 | 1.07 | 0.41 |
| | $10^3$ | 2.90 | 0.98 | 1.15 | 1.00 | 2.90 | 32.0 | 0.84 | 0.43 |
| | $10^4$ | 2.90 | 0.99 | 1.26 | 1.02 | 2.92 | 336.6 | 0.19 | 0.35 |
| 10 | $10^2$ | 2.90 | 10.13 | 1.04 | 9.24 | 2.89 | 32.9 | 1.06 | 2.21 |
| | $10^3$ | 2.91 | 10.02 | 1.12 | 10.01 | 2.89 | 337.8 | 0.75 | 1.30 |
| | $10^4$ | 2.91 | 10.01 | 1.34 | 9.72 | 2.90 | 3305.6 | 0.22 | 1.98 |

Table 1 summarizes the test MSE results and Table 2 (left) summarizes the estimated variance components results. As can be seen LMMNN reaches the smallest test MSE on average and with a considerable gap from the other methods, when standard errors are taken into account. This is particularly true when RE variance $\sigma_b^2$ and cardinality $q$ are high. As for the estimated variance components $\hat{\sigma}_e^2, \hat{\sigma}_b^2$, LMMNN reaches a good estimation for both, while R's lme4 reaches a poor estimation for $\sigma_e^2$ without adding appropriate non-linear and interaction terms, resulting in worse prediction performance. We also plot in additional material Figure 1 the predicted random effects $\hat{b}$ vs. the true random effects $b$ for different $q$s, the category size distribution and also $\hat{y}_{te}$ vs. $y_{te}$. Finally Table 1 in additional material summarizes mean runtime and number of epochs.

A more challenging scenario is when $g$ is some linear transformation $W_{q \times d}$ of $Z$ into a lower dimension $d$:

$$y = (X_1 + \cdots + X_{10}) \cdot \cos(X_1 + \cdots + X_{10}) + 2 \cdot X_1 \cdot X_2 + ZWb + \epsilon, \qquad (16)$$

where we use $d = 0.1 \cdot q$ and sample $W$ from a $\mathbb{U}(-1, 1)$ distribution. $b$ is now a $d$-length vector of random effects, sampled from a $\mathbb{N}(0, \sigma_b^2)$ distribution ($\sigma_b^2 \in \{0.1, 1, 10\}$). We use a Keras Embedding layer on $Z$ to learn $W$, and notice now we cannot avoid inversion of $V$ when calculating the loss in (8) or in predicting $\hat{b}$ in (5). $V$ is a $N \times N$ matrix, so it is perfectly reasonable to invert it for a batch size of 30, but for predicting $\hat{b}$ on the entire training set at the end of training this means creating and inverting a matrix of dimensions 80,000 $\times$ 80,000. In practice we sample 10,000 observations from the training set for estimating $V$ and this seems to work well, as summarized by Table 3 which shows the test MSE scores for all methods. Our method performs the best especially at high cardinality $q$ where the reduction in MSE can go over 80%. We do note that LMMNN struggles

Table 3: Simulated model with $g(Z) = ZW$, mean test MSEs and standard errors in parentheses. Bold results are non-inferior to the best result in a paired t-test.

| $\sigma_b^2$ | $q$ | Ignore | OHE | Embeddings | lme4 | MeNets | LMMNN |
|---|---|---|---|---|---|---|---|
| 0.1 | $10^2$ | 1.42 (.039) | 1.22 (.018) | 1.19 (.024) | 2.91 (.021) | 1.25 (.084) | **1.13 (.013)** |
| | $10^3$ | 4.92 (.345) | 1.49 (.033) | 1.43 (.035) | 2.95 (.020) | 1.44 (.061) | **1.16 (.013)** |
| | $10^4$ | 35.1 (.456) | 3.25 (.086) | 3.39 (.116) | 3.42 (.036) | 7.35 (1.9) | **1.59 (.023)** |
| 1 | $10^2$ | 4.13 (.626) | 1.32 (.025) | 1.31 (.033) | 2.88 (.023) | 1.40 (.106) | **1.16 (.011)** |
| | $10^3$ | 35.6 (2.78) | 2.49 (.151) | 2.56 (.355) | 2.96 (.045) | 7.00 (1.9) | **1.19 (.028)** |
| | $10^4$ | 334 (18) | 9.13 (2.29) | 14.4 (2.89) | 4.29 (.1) | 143.3 (32) | **3.43 (.757)** |
| 10 | $10^2$ | 32.5 (6.86) | 1.74 (.121) | 2.43 (.317) | 2.90 (.023) | 12.0 (3.03) | **1.12 (.012)** |
| | $10^3$ | 324 (25) | 8.94 (.79) | 9.27 (.92) | 2.96 (.031) | 164 (18) | **1.20 (.020)** |
| | $10^4$ | 3337 (134) | 60.1 (4.6) | 91.1 (4.5) | **13.8 (1.2)** | 2880 (463) | **13.3 (2.0)** |

in finding good $\hat{\sigma}_e^2, \hat{\sigma}_b^2$ estimates in such a complex scenario, as summarized in Table 2 (right), but its estimates are still substantially better than those of R's lme4.

## 5.2 Real Data

We first describe the datasets and prediction tasks used in this work.

**Estimating physical activity from self-reported behaviors from the UK Biobank**  The UK Biobank is an ongoing large scale cohort study where over 500,000 individuals across the UK aged 40-69 were first invited to 22 assessment centers in 2006-2010, surveyed and measured for various behaviors and metrics, such as gender, job, personal habits, current and historical health status [20]. Participants' information has been de-identified and strict protocols exist for obtaining and disposing of the UK Biobank data. Several sub-samples of this cohort have also been invited to participate in more expensive and time consuming surveys such as neuroimaging and genome sequencing. One sub-sample of over 100,000 participants wore a wrist-mounted accelerometer for a period of 7 days [7]. We follow Pearce et al. [19] who attempted to predict these accelerometer-based physical activity (PA) data from much more readily available 14 self-reported behaviors such as no. of hours watching TV and sleeping, and gender. The motivation behind doing so is that physical activity is a strong marker for diseases outcomes such as cancer and respiratory disease mortality, yet it is hard to measure. The authors used linear regression and ignored some important features. Of particular interest here is the current job feature, a categorical feature with over 300 levels, unevenly distributed across participants. We attempt to improve prediction of physical activity by adding the job feature and using DNN in a LMMNN approach.

**Estimating Drugs 1-10 rating from textual reviews from Drugs.com**  The website drugs.com is an online pharmaceutical database of drugs, providing information, reviews and ratings of thousands of doctor prescribed drugs. Gräßer et al. [10] scraped drugs.com for over 215,000 anonymized reviews and ratings which are available in the UCI Machine Learning Repository [8]. The authors discretized the 1-10 ratings into "negative", "neutral" and "positive" sentiments and used logistic regression on the text reviews processed to n-grams, to predict sentiment in a classification framework. Here we predict the rating itself in a regression framework using DNN, with a standard word embeddings layer followed by a LSTM layer and a single neuron output. A high-cardinality categorical feature which should improve prediction is the drug itself and there are over 3,600 drugs reviewed and rated in this dataset (so each drug has about 60 reviews on average, but these are unevenly distributed).

**Landmark localization in CelebA facial images**  The CelebA dataset (Liu et al. [17]) contains 202,599 cropped facial images from 10,177 celebrities, where each celebrity has between 1 and 35 images, annotated for various attributes and landmarks such as the location of the tip of the nose and mouth. We use two convolutional neural networks to predict the $(X, Y)$ location of the nose and expect that treating facial images from the same individual as repeated measures should improve prediction. Therefore, the celebrity identity itself is the RE feature used here in the LMMNN framework.

Table 4: Real data features summary table

| Dataset | $n$ | $q$ | $p$ | Categorical | $y$ | Input Type | DNN |
|---------|-----|-----|-----|-------------|-----|------------|-----|
| UKB PA | 96K | 350 | 15 | job | PA | Tabular | MLP |
| Drugs | 215K | 3.6K | 10K | drug | rating | Text | LSTM |
| CelebA | 202K | 10K | 218x178x3 | identity | noseX-Y | Images | CNN |
| Airbnb | 50K | 40K | 196 | host | log(price) | Tabular | MLP |

Table 5: Real data 5-CV mean test MSEs and standard errors in parentheses. Bold results are non-inferior to the best result in a paired t-test.

| Dataset | Ignore | OHE | Embeddings | MeNets | LMMNN |
|---------|--------|-----|------------|--------|-------|
| UKB PA | **0.812 (.008)** | 0.816 (.009) | 0.817 (.010) | **0.811 (.009)** | **0.809 (.008)** |
| Drugs | 2.74 (.032) | 2.77 (.005) | 2.72 (.051) | 2.81 (.031) | **2.66 (.006)** |
| CelebA noseX | **1.68 (.05)** | – | 3.6 (.3) | 7.6 (.3) | **1.54 (.07)** |
| CelebA noseY | 1.64 (.09) | – | 2.5 (.2) | 12.3 (1.1) | **1.39 (.04)** |
| Airbnb | 0.156 (.002) | – | 0.158 (.003) | 0.153 (.003) | **0.142 (.002)** |

**Predicting prices of Airbnb rentals**   The website insideairbnb.com scrapes publicly available data from Airbnb rentals under the CC0 1.0 license [5]. Kalehbasti et al. [15] attempted to predict the log price of a rental from various features such as neighborhood, number of rooms and textual features extracted from reviews, overall 755 features. After performing variable selection they used multiple methods including DNN and found support vector regression (SVR) to give the best result in terms of test MSE. Their dataset holds about 50,000 listings with over 39,000 hosts which they did not even consider when performing variable selection. Some hosts publish only 1 rental, while some over 100.

In each dataset we use the same baseline DNN architecture detailed in the original analysis, if available. If the original analysis did not include the use of DNNs or did not provide specific details, we use a standard architecture such as the LSTM used for modeling the Drugs dataset reviews. The RE variable is either ignored, goes through embeddings or OHE (if possible computationally) before entering the network, used in the MeNets framework or used in the LMMNN framework, where we use $g(Z) = Z$ always. For specifics on baseline network architecture for each dataset see additional material. We perform 5-fold cross validation where in each iteration we use 10% of the train set (which is 80% of the entire data) as validation data, and train the network until no improvement in validation loss is seen within 10 epochs. All runs are made on Google Colab Pro, with a Nvidia Tesla P100 GPU.

Table 4 describes the datasets in general, Table 5 summarizes the test MSE results and Table 3 in additional material summarizes mean runtime and number of epochs. It can be seen that including the high-cardinality variable in each of these datasets within a LMMNN framework achieves the lowest test MSE. We also refer the reader to additional material Figure 2 where we show true test $y$ vs. predicted $y$ scatter plots. Comparing to the results of the original papers is challenging since not all have supplied data and methods in a fully reproducible way and they use different metrics. Notwithstanding, we can verify our results are at least comparable in performance: For estimating physical activity in the UK Biobank, Pearce et al. [19] report achieving a test $R^2$ of between 14 and 17%, while LMMNN achieves on average 19% (higher is better). For estimating drugs rating from text reviews we cannot compare to Gräßer et al. [10] results as they worked in a classification setting. For localizing the nose in the CelebA facial images, we are not aware of any MSE benchmarks but note that the LMMNN result of less than 2 pixels on average seems satisfactory. Finally, for the Airbnb rental price prediction, Kalehbasti et al. [15] report achieving a test MSE measure of 0.157 for a neural network, while LMMNN achieves on average a mean test MSE of 0.142, which is even better than the best result they achieved with SVR, 0.147.

## 6   LMM Extensions

The current paper is focused on regression settings with a single categorical feature, e.g. the subject with repeated measurements, as we wanted to take advantage of the computationally attractive LMM

Table 6: Simulated model with $g(Z) = Z$ and two categorical features, mean test MSEs and standard errors in parentheses. Bold results are non-inferior to the best result in a paired t-test.

| $\sigma_{b1}^2$ | $\sigma_{b2}^2$ | $q_1$ | $q_2$ | Ignore | OHE | Embeddings | lme4 | LMMNN |
|---|---|---|---|---|---|---|---|---|
| 0.5 | 0.5 | $10^3$ | $10^3$ | 2.18 (.03) | 1.45 (.02) | 1.34 (.01) | 2.97 (.03) | **1.13 (.01)** |
| | | $10^3$ | $10^4$ | 2.15 (.02) | 1.70 (.01) | 1.68 (.02) | 3.12 (.03) | **1.23 (.01)** |
| | | $10^4$ | $10^4$ | 2.13 (.02) | 1.83 (.02) | 1.80 (.02) | 3.23 (.03) | **1.30 (.00)** |
| 0.5 | 5.0 | $10^3$ | $10^3$ | 6.73 (.04) | 1.66 (.03) | 1.57 (.02) | 3.00 (.04) | **1.12 (.00)** |
| | | $10^3$ | $10^4$ | 6.75 (.04) | 2.20 (.03) | 2.01 (.03) | 3.31 (.02) | **1.29 (.01)** |
| | | $10^4$ | $10^3$ | 6.50 (.05) | 1.88 (.03) | 1.92 (.04) | 3.15 (.01) | **1.23 (.01)** |
| | | $10^4$ | $10^4$ | 6.68 (.12) | 2.48 (.03) | 2.16 (.02) | 3.43 (.01) | **1.37 (.01)** |
| 5.0 | 5.0 | $10^3$ | $10^3$ | 11.26 (.19) | 1.83 (.02) | 1.80 (.07) | 2.97 (.02) | **1.14 (.02)** |
| | | $10^3$ | $10^4$ | 11.33 (.19) | 2.36 (.03) | 2.11 (.02) | 3.32 (.02) | **1.30 (.01)** |
| | | $10^4$ | $10^4$ | 11.24 (.09) | 3.02 (.03) | 2.55 (.000) | 3.69 (.02) | **1.49 (.02)** |

feature described in equation (6) of avoiding any matrix inversions in calculating the predicted RE and the NLL loss. However, in further experimenting with LMMNN we note that it seems to perform well with more complex mixed effects scenarios which do necessitate inverting *part* of the $V$ covariance matrix in equation (8) on each mini-batch during training. Apart from the simulation in which we defined $g(Z) = ZW$ and used embeddings in the network architecture, other scenarios include having multiple categorical features, the *random slopes* model which adds a "slope" random effect to each categorical level, and even kriging over random fields, where $V$ is assumed to contain spatio-temporal dependencies.

To give the reader a flavour for how LMMNN can be generalized to more scenarios, consider a situation of two uncorrelated categorical features, having $q_1$ and $q_2$ levels. Now $Z$ in (7) is an indicator matrix of size $n \times (q_1 + q_2)$, $b$ of size $q_1 + q_2$ distributes as before $\mathbb{N}(0, D)$, only now $D$ is of size $(q_1 + q_2) \times (q_1 + q_2)$, and has structure:

$$D = \begin{bmatrix} \sigma_{b1}^2 I_{q_1} & 0_{q_1 \times q_2} \\ 0_{q_2 \times q_1} & \sigma_{b2}^2 I_{q_2} \end{bmatrix}, \tag{17}$$

Now $\psi$ holds three variance components $[\sigma_e^2, \sigma_{b1}^2, \sigma_{b2}^2]$, and we can write the NLL and its derivatives in (8) and (9) exactly as before. However, notice that $V$ in this case is not necessarily block-diagonal anymore, and LMMNN calculates the inverse on part of it for each mini-batch.

We show in Table 6 the mean test MSE results of a simulation similar to the first simulation in Section 5.1, where $g$ is the identity matrix, $f$ is in (15), $\sigma_e^2 = 1$, $n = 100,000$, using the same DNN architectures and rules. In fact the only difference from the simulation in Section 5.1 is that now we have *two* high-cardinality uncorrelated categorical features, we vary $q_1, q_2, \sigma_{b1}^2, \sigma_{b2}^2$ and LMMNN now has to estimate *three* variance components. As can be seen, LMMNN's performance is superior to all other methods. It also estimates the variance components quite accurately when compared to R's lme4, with a certain cost in runtime, see Tables 4 and 5 in additional material.

## 7 Conclusion

In this work we propose to treat high-cardinality categorical features in DNN as RE variables, to avoid the possible pitfalls of overfitting, over-parameterization and scalability issues often encountered when opting to use OHE or entity embeddings. LMMNN introduces a novel negative log likelihood loss function inspired by the well researched linear mixed effects model and its positive effect on DNN prediction performance is shown, with a simple implementation in Keras which can be plugged in to almost any regression DNN architecture, and is freely available on Github.

Treating categorical features with many levels in DNN as random effects seems to be promising as our simulated and real data results show and we intend to further pursue this direction in future work.

## Acknowledgments and Disclosure of Funding

This study was supported in part by a fellowship from the Edmond J. Safra Center for Bioinformatics at Tel-Aviv University, and by Israeli Science Foundation grant 1804/16. UK Biobank research has been conducted using the UK Biobank Resource under Application Number 56885.

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
