# Using Random Effects to Account for High-Cardinality Categorical Features and Repeated Measures in Deep Neural Networks

**Giora Simchoni**
Department of Statistics
Tel Aviv University
Tel Aviv, Israel, 69978
gsimchoni@tauex.tau.ac.il

**Saharon Rosset**
Department of Statistics
Tel Aviv University
Tel Aviv, Israel, 69978
saharon@tauex.tau.ac.il

## 1 Additional table and figures

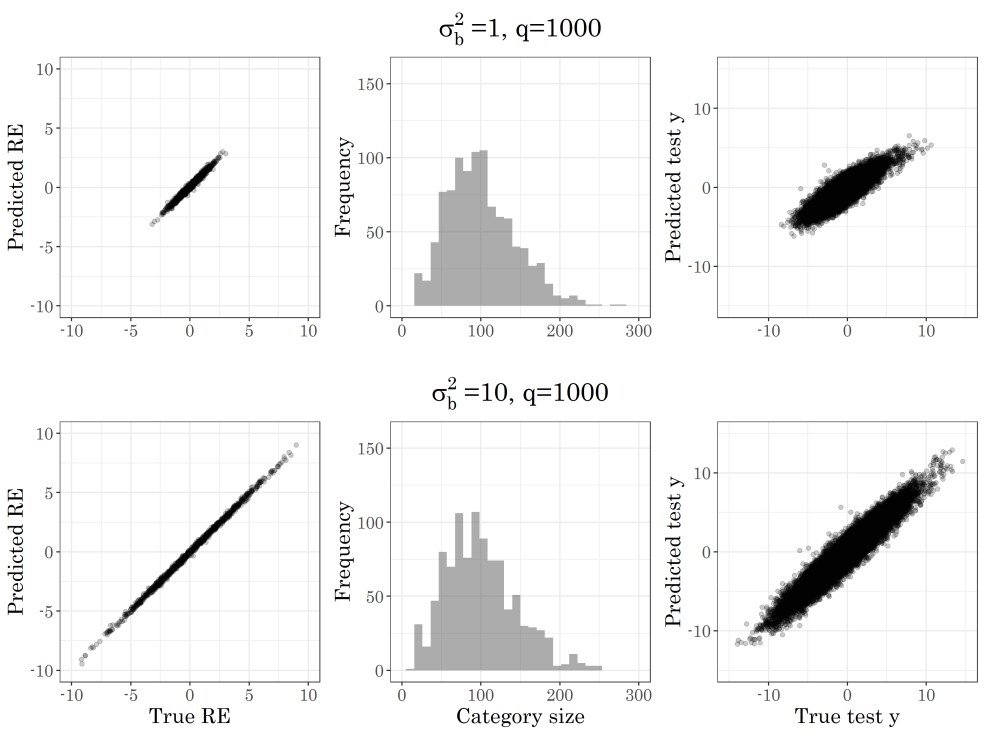

Figure 1: Simulation results when $g(Z) = Z, q = 1000, \sigma_b^2 = 1$ (top) and $\sigma_b^2 = 10$ (bottom)

35th Conference on Neural Information Processing Systems (NeurIPS 2021).

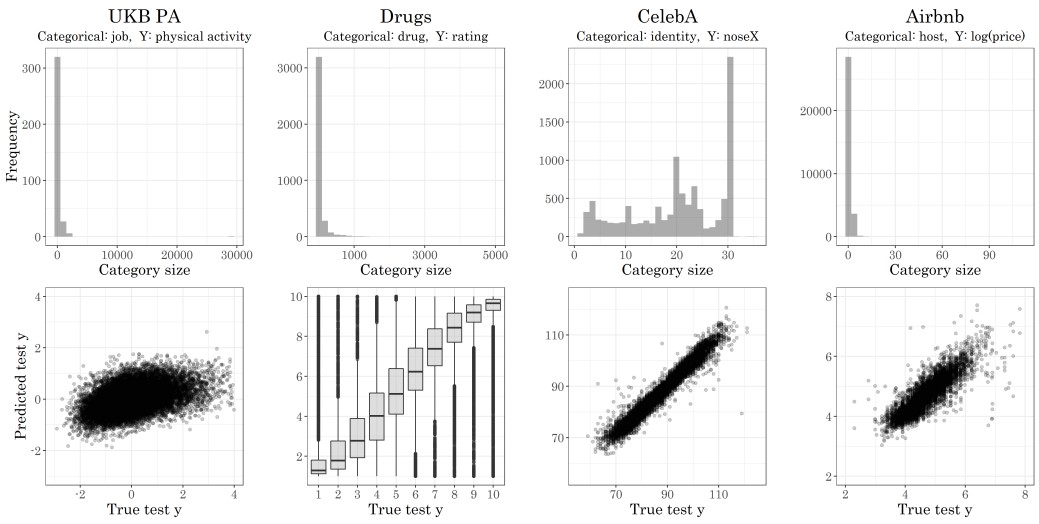

Figure 2: Real data predicted vs. true results and category size distribution

Table 1: Simulated model with $g(Z) = Z$, mean runtime (minutes) and number of epochs in parentheses.

| $\sigma_b^2$ | $q$ | Ignore | OHE | Embeddings | lme4 | MeNets | LMMNN |
|---|---|---|---|---|---|---|---|
| 0.1 | $10^2$ | 2.9 (32) | 2.9 (27) | 3.9 (28) | 0.01 (–) | 26.4 (96) | 7.3 (36) |
| | $10^3$ | 3.1 (35) | 1.4 (15) | 2.2 (16) | 0.01 (–) | 48.3 (259) | 5 (33) |
| | $10^4$ | 3.0 (35) | 2.3 (12) | 1.7 (12) | 0.02 (–) | 47.5 (275) | 6.8 (42) |
| 1 | $10^2$ | 2.9 (33) | 3.2 (35) | 4.2 (31) | 0.01 (–) | 21.2 (82) | 6 (40) |
| | $10^3$ | 1.4 (16) | 1.9 (20) | 3.2 (23) | 0.01 (–) | 79.3 (434) | 4.9 (32) |
| | $10^4$ | 2.2 (25) | 2.5 (14) | 2 (14) | 0.02 (–) | 51.1 (300) | 5.8 (36) |
| 10 | $10^2$ | 2.1 (24) | 1.6 (18) | 3.3 (25) | 0.01 (–) | 17.6 (65) | 5.4 (36) |
| | $10^3$ | 2.4 (27) | 1.6 (17) | 3.4 (25) | 0.01 (–) | 34.5 (196) | 6 (39) |
| | $10^4$ | 2.6 (29) | 2.9 (16) | 2.6 (18) | 0.02 (–) | 50.9 (300) | 5.9 (37) |

Table 2: Simulated model with $g(Z) = ZW$, mean runtime (minutes) and number of epochs in parentheses.

| $\sigma_b^2$ | $q$ | Ignore | OHE | Embeddings | lme4 | MeNets | LMMNN |
|---|---|---|---|---|---|---|---|
| 0.1 | $10^2$ | 3.4 (38) | 2.6 (29) | 4.1 (31) | 0.01 (–) | 13.3 (63) | 20.7 (127) |
| | $10^3$ | 2.7 (31) | 2 (21) | 3.7 (28) | 0.01 (–) | 44.1 (279) | 24.4 (141) |
| | $10^4$ | 2.9 (32) | 3.5 (20) | 4 (29) | 0.02 (–) | 54.9 (300) | 442.8 (101) |
| 1 | $10^2$ | 3.1 (29) | 3.3 (30) | 3.8 (24) | 0.01 (–) | 15.3 (76) | 56.6 (291) |
| | $10^3$ | 2.8 (31) | 2.5 (26) | 4.2 (32) | 0.01 (–) | 23.9 (148) | 50.8 (286) |
| | $10^4$ | 2.5 (28) | 6.3 (38) | 4.5 (33) | 0.02 (–) | 27.1 (146) | 832.1 (200) |
| 10 | $10^2$ | 2.2 (25) | 3.2 (36) | 4 (32) | 0.01 (–) | 10.8 (55) | 84.2 (500) |
| | $10^3$ | 2.9 (33) | 1.2 (12) | 3.5 (26) | 0.01 (–) | 2.9 (17) | 87.2 (500) |
| | $10^4$ | 2.5 (19) | 7.9 (37) | 8.1 (40) | 0.02 (–) | 6 (32) | 1191 (250) |

Table 3: Real data 5-CV mean runtime (minutes) and number of epochs in parentheses.

| Dataset | Ignore | OHE | Embeddings | MeNets | LMMNN |
|---|---|---|---|---|---|
| UKB PA | 3.0 (45) | 1.4 (20) | 2.3 (24) | 132.2 (26) | 4.9 (34) |
| Drugs | 30.4 (33) | 34.6 (38) | 36.8 (40) | 58.5 (45) | 62.6 (68) |
| CelebA noseX | 335 (33) | – | 430 (39) | 576 (21) | 334 (31) |
| CelebA noseY | 317 (27) | – | 325 (27) | 496 (17) | 282 (23) |
| Airbnb | 0.5 (102) | – | 13.8 (25) | 135.2 (500) | 1.6 (20) |

Table 4: Simulated model with $g(Z) = Z$ and two categorical features, estimated variance components on average

| | | | | lme4 | | | LMMNN | | |
|---|---|---|---|---|---|---|---|---|---|
| $\sigma_{b1}^2$ | $\sigma_{b2}^2$ | $q_1$ | $q_2$ | $\hat{\sigma}_e^2$ | $\hat{\sigma}_{b1}^2$ | $\hat{\sigma}_{b2}^2$ | $\hat{\sigma}_e^2$ | $\hat{\sigma}_{b1}^2$ | $\hat{\sigma}_{b2}^2$ |
| 0.5 | 0.5 | $10^3$ | $10^3$ | 2.90 | 0.52 | 0.48 | 1.13 | 0.50 | 0.53 |
| | | $10^3$ | $10^4$ | 2.92 | 0.50 | 0.48 | 1.13 | 0.51 | 0.50 |
| | | $10^4$ | $10^4$ | 2.90 | 0.50 | 0.49 | 1.13 | 0.50 | 0.50 |
| 0.5 | 5.0 | $10^3$ | $10^3$ | 2.89 | 0.51 | 4.98 | 1.12 | 0.49 | 5.04 |
| | | $10^3$ | $10^4$ | 2.92 | 0.50 | 5.01 | 1.13 | 0.51 | 4.98 |
| | | $10^4$ | $10^3$ | 2.88 | 0.50 | 4.94 | 1.14 | 0.52 | 4.78 |
| | | $10^4$ | $10^4$ | 2.91 | 0.50 | 4.97 | 1.14 | 0.49 | 5.01 |
| 5.0 | 5.0 | $10^3$ | $10^3$ | 2.91 | 5.14 | 4.98 | 1.12 | 5.02 | 4.99 |
| | | $10^3$ | $10^4$ | 2.90 | 5.03 | 4.95 | 1.14 | 5.11 | 5.00 |
| | | $10^4$ | $10^4$ | 2.91 | 5.05 | 4.96 | 1.14 | 4.90 | 5.00 |

Table 5: Simulated model with $g(Z) = Z$ and two categorical features, mean runtime (minutes) and number of epochs in parentheses.

| $\sigma_{b1}^2$ | $\sigma_{b2}^2$ | $q_1$ | $q_2$ | Ignore | OHE | Embeddings | lme4 | LMMNN |
|---|---|---|---|---|---|---|---|---|
| 0.5 | 0.5 | $10^3$ | $10^3$ | 2.3 (26) | 1.7 (16) | 2.7 (19) | 0.04 (–) | 6.2 (37) |
| | | $10^3$ | $10^4$ | 2.4 (27) | 2.7 (14) | 2.1 (14) | 0.04 (–) | 7.8 (42) |
| | | $10^4$ | $10^4$ | 3.4 (39) | 4.5 (14) | 2.1 (13) | 0.04 (–) | 10.9 (43) |
| 0.5 | 5.0 | $10^3$ | $10^3$ | 2.5 (28) | 2.2 (20) | 3.4 (24) | 0.04 (–) | 5.7 (34) |
| | | $10^3$ | $10^4$ | 2.2 (25) | 3.4 (18) | 2.7 (17) | 0.04 (–) | 6.5 (34) |
| | | $10^4$ | $10^3$ | 2.4 (28) | 3.2 (17) | 2.4 (16) | 0.04 (–) | 6.6 (35) |
| | | $10^4$ | $10^4$ | 2.1 (23) | 4.7 (15) | 2.5 (16) | 0.04 (–) | 8.8 (33) |
| 5.0 | 5.0 | $10^3$ | $10^3$ | 1.9 (22) | 3.4 (32) | 3.9 (27) | 0.03 (–) | 5.6 (33) |
| | | $10^3$ | $10^4$ | 2.1 (24) | 3.5 (19) | 2.9 (19) | 0.04 (–) | 7.4 (40) |
| | | $10^4$ | $10^4$ | 2.4 (27) | 4.8 (15) | 3.4 (22) | 0.04 (–) | 9.3 (36) |

# 2  Simulations

## 2.1  $g(Z) = Z$

- This simulation implements equation (15) from the paper
- All runs were made on a Nvidia Quadro P620 GPU on a Windows machine, implemented in Python 3.8 Numpy + Pandas suite, Keras and Tensorflow
- Code is fully available in the `lmmnn` package on Github
- Running code: see details in package `README` file

- $n = 100,000$, $\sigma_e^2 = 1$
- $\epsilon \sim \mathbb{N}\left(0, \sigma_e^2\right)$
- $p = 10$, i.e. 10 fixed predictors in $X$ where each $X_k \sim \mathbb{U}\left(-1, 1\right)$, $k \in \{1, \ldots p\}$
- Single categorical random variable with $q$ levels where $q \in \{100, 1000, 10000\}$
- $b$ is a $q$-length vector of i.i.d random effects, sampled from a $\mathbb{N}\left(0, \sigma_b^2\right)$ distribution where $\sigma_b^2 \in \{0.1, 1, 10\}$
- $n_{iter} = 5$ replications for each $(q, \sigma_b^2)$ combination
- In total $3 \times 3 \times 5 = 45$ runs, for each of 6 types: {Ignore, OHE, Embedding, lme4, MeNets, LMMNN}
- At each run 80% (80,000) of the simulated data is used as training set, of which 10% (8,000) is used as validation set which the network only uses to check for early stopping. That leaves 20% of the data (20,000) as testing set and we record $y_{te}$ vs. $\hat{y}_{te}$ mean squared error and standard error over 10 replication per condition.
- Max no. of epochs: 500
- batch size: 30
- We use a Keras `EarlyStopping` callback to stop training if no improvement has been observed for 10 epochs.
- Baseline DNN architecture: 4 hidden fully connected layers with {100, 50, 25, 12} number of neurons, a ReLU activation and a Dropout of 25% in each, and a final output layer with a single neuron with no activation
- Loss used is mean squared error in all networks except for LMMNN
- Optimizer: Adam, Keras default params
- Specific architecture:
    - Ignore: input is $X$ of dimension $p$
    - OHE: input is $X$ and $Z$ of dimension $p + q$
    - Embedding: input is $X$ of dimension $p$, in addition $Z$ goes through an `Embedding` layer which maps $q$ levels to a $d = 0.1 \cdot q$ vector, so input dimension is $p + d$
    - LMMNN: input is $X$ of dimension $p$, in addition $Z$, $y_{tr}$ and final single neuron output are input to the custom `NLL` loss layer. We initialize $(\hat{\sigma}_e^2, \hat{\sigma}_b^2)$ to be (1.0, 1.0). Note that $Z$ is input to the `NLL` loss layer via a sparse vector, we do not actually keep a $n \times q$ matrix.
    - MeNets: input is $X$ of dimension $p$ and the layer before last of 12 neurons is the feature mapping used in the MeNets V-EM algorithm.

## 2.2 $g(Z) = ZW$

- This simulation implements equation (16) from the paper
- Changes from simulation in 2.1:
    - $W_{q \times d}$ is a linear transformation of $Z$
    - $d = 0.1 \cdot q$ and we sample $W$ from a $\mathbb{U}\left(-1, 1\right)$ distribution
    - $b$ is now a $d$-length vector of i.i.d random effects, sampled from a $\mathbb{N}\left(0, \sigma_b^2\right)$ as before
    - Max no. of epochs when $q = 10,000$: 250
    - DNNs architecture remains the same except in LMMNN where $Z$ goes through a `Embedding` layer which maps $q$ levels to a $d = 0.1 \cdot q$ vector before it is input to the `NLL` loss layer
    - For LMMNN we use our own custom `EarlyStoppingWithSigmasConvergence` callback which also makes sure the estimated variance components $(\hat{\sigma}_e^2, \hat{\sigma}_b^2)$ have converged for 10 epochs.

## 2.3 $g(Z) = Z$ with two categorical features

- This simulation implements equation (15) from the paper with *two* categorical features as explained in Section 6.
- Changes from simulation in 2.1:
    - $\sigma_{b1}^2$ and $\sigma_{b2}^2$ are varied in $\{0.5, 5.0\}$, $q_1$ and $q_2$ are varied in $\{1,000, 10,000\}$ with all possible 10 unique combinations
    - In total $10 \times 5 = 50$ runs, for each of 5 types: {Ignore, OHE, Embedding, lme4, LMMNN}
    - DNNs architecture remains the same: Ignore means ignoring both categorical features, OHE means one-hot encoding both categorical features, embedding means embedding each of the categorical features to its own vector using two Keras Embedding layers

# 3 Real Data

## 3.1 UKB-PA: Estimating physical activity from self-reported behaviors from the UK Biobank

- Relevant notebook: `ukb_pa.ipynb`
- Data availability: upon request from the UK Biobank only
- Physical activity (PA) definition: Subjects wore an accelerometer on their wrist for 7 days. Physical activity is measured in ENMO units (euclidean norm minus one), calculated on the acceleration vector in three axes, and negative values were truncated to zero. Mean wrist ENMO in m-g was summarised across valid wear-time.
- ETL: We follow instructions by Pearce et al. (2020), implemented in R. At high level, we filter out subjects wearing the accelerometer for less than 72 hours or having ENMO of over 80. Each of the categorical behavioral variables e.g. "frequency of walking for pleasure" is converted to numerical with a simple mapping e.g. "once a week" is converted to 1 and "every day" is converted to 7. Finally the PA dependent variable is standardized to have a mean of 0 and standard deviation of 1 for each fold, for the training set.
- $n = 96,629$
- Categorical feature: Job ($q = 353$)
- Fixed features:
    - Gender
    - Heavy work time
    - Walking during work time
    - Sedentary during work time
    - Moderate-to-vigorous physical activity time (MVPA)
    - Walking for pleasure time
    - Strenuous sport time
    - Other activities time
    - Light DIY time
    - Heavy DIY time
    - TV time
    - Computer time
    - Sleep time
    - Getting about method (OHE): walk, cycle, transport, other
    - Commute method (OHE): walk, cycle, transport, other
- Baseline DNN architecture: Pearce et al. did not use DNNs, but two separate linear regressions, for men and women. We use a simple MLP of 2 fully connected layers with ReLU activation of 10 and 5 neurons, followed by a single output neuron with no activation.
- Split policy: 5-fold CV, from each training fold 10% of data is kept as validation data which the network uses for early stopping

- Max no. of epochs: 2000
- Batch size: 30
- Callbacks: EarlyStopping with patience = 10
- Runtime: Google Colab, Nvidia Tesla P100 GPU

### 3.2 Drugs: Estimating Drugs 1-10 rating from textual reviews from Drugs.com

- Relevant notebook: `drugs.ipynb`
- Data availability: freely available in the UCI Machine Learning Repository `https://archive.ics.uci.edu/ml/datasets/Drug+Review+Dataset+%28Drugs.com%29`
- ETL: We bind Gräßer et al. (2018) training and testing set into a single set, on which we perform regular 5-fold cross validation. We use only the drugs reviews, perform standard tokenization to words using Keras `Tokenizer` with maximum 10,000 most common words (which is in a sense the fixed feature dimension $p$) and each review is cut at maximum length 100 words.
- $n = 215,063$
- Categorical feature: Drug ($q = 3,671$)
- Baseline DNN architecture: Gräßer et al. did not use DNNs, we therefore use a standard text input architecture. After tokenization reviews are fed into a standard Embedding layer of dimension 100. We then use a LSTM layer of 64 kernels, followed by a single output neuron with no activation.
- Split policy: 5-fold CV, from each training fold 10% of data is kept as validation data which the network uses for early stopping
- Max no. of epochs: 100
- Batch size: 30
- Callbacks: EarlyStopping with patience = 5
- Runtime: Google Colab, Nvidia Tesla P100 GPU

### 3.3 Airbnb: Predicting prices of Airbnb rentals

- Relevant notebook: `airbnb.ipynb`
- Data availability: freely available following Kalehbasti et al. (2019) instructions at `https://github.com/PouyaREZ/AirBnbPricePrediction`
- ETL: We run Kalehbasti et al. code as is including their variable selection procedure and log transformation for the price. We then bind their training, validation and testing into a single set, on which we perform regular 5-fold cross validation.
- $n = 49,976$
- Categorical feature: Host ($q = 39,393$)
- Fixed features: $p = 196$ features after variable selection, e.g.: does rental has 24-hour check-in, does it have a dryer, is it kids friendly etc.
- Baseline DNN architecture: We use Kalehbasti et al. architecture: a simple MLP of 2 fully connected layers with ReLU activation of 20 and 5 neurons, followed by a single output neuron with no activation. We also use their parameters for the Adam optimizer.
- Split policy: 5-fold CV, from each training fold 10% of data is kept as validation data which the network uses for early stopping
- Max no. of epochs: 500
- Batch size: 30
- Callbacks: EarlyStopping with patience = 10
- Runtime: Google Colab, Nvidia Tesla P100 GPU

### 3.4 CelebA: Localizing the tip of the nose in facial images

- Relevant notebook: `celeba.ipynb`
- Data availability: freely available from Liu et al. (2015) at `http://mmlab.ie.cuhk.edu.hk/projects/CelebA.html`
- ETL: None
- $n = 202,599$
- Categorical feature: Identity ($q = 10,177$)
- Baseline DNN architecture: Liu et al. (2015) did not use a CNN to localize any of the landmarks. We found that a standard CNN architecture works fine:
  - Conv. 2D layer, 32 kernels, (5, 5) strides, padding valid, ReLU activation
  - Max 2D pooling, (2, 2) pool size
  - Conv. 2D layer, 64 kernels, (5, 5) strides, padding valid, ReLU activation
  - Max 2D pooling, (2, 2) pool size
  - Conv. 2D layer, 32 kernels, (5, 5) strides, padding valid, ReLU activation
  - Max 2D pooling, (2, 2) pool size
  - Conv. 2D layer, 16 kernels, (5, 5) strides, padding valid, ReLU activation
  - Max 2D pooling, (2, 2) pool size
  - Flatten, Dropout of 50
  - Fully connected layer of 100 neurons and ReLU activation
  - Single output neuron with no activation
- Split policy: 5-fold CV, from each training fold 10% of data is kept as validation data which the network uses for early stopping,
- Max no. of epochs: 100
- Batch size: 20
- Callbacks: EarlyStopping with patience = 10
- Runtime: Google Colab, Nvidia Tesla P100 GPU