# OpenReview forum: "Using Random Effects to Account for High-Cardinality Categorical Features and Repeated Measures in Deep Neural Networks"
_NeurIPS.cc/2021/Conference — NeurIPS 2021 Poster_

### Official Review · Reviewer_avbJ · 2021-07-15

**Rating:** 6
**Confidence:** 4

**Summary:**

The logic of this paper is clear and the experiment is complete, but it can be further improved.

**Limitations And Societal Impact:**

Yes

**Main Review:**

By considering the problem of high-cardinality categorical features are hard to scale, require much space, and are hard to interpret or may overfit the data, this paper proposes the LMMNN model, which adopting the corresponding negative log-likelihood loss from the linear mixed models statistical literature and integrates it in a deep learning framework. They test their model on simulated as well as real datasets with a single categorical feature with high cardinality, using various baselines neural networks architectures such as CNN and LSTM, and various applications in e-commerce, healthcare, and computer vision. The experiments on the real datasets demonstrate the application value of their model. In addition, the logic of the paper is clear and the supplementary materials are complete. All in all, this is a meaningful study.

However, I still have some concerns as follows:

1.	Line 14: convolutional networks -> convolutional neural networks (CNN)
2.	As far as we know, complex DNN generally have better feature screening capabilities, but the DNN they used as baseline models were relatively simple. How the authors demonstrate that their method is superior to more advanced network architectures in improving performance.
3.	The authors should improve their language writing to avoid grammar mistakes.


**Time Spent Reviewing:**

2 days

---

> ### Author Response · Authors · 2021-08-10
> **Comment to Reviewer avbJ**
>
> Thank you for your thoughtful review.
>
> * "DNN they used as baseline models were relatively simple" – we agree, the purpose was to show how LMMNN could be plugged in to various architectures and we felt it is proper to “start from the basics”. The interesting comparison in our opinion is to other existing methods of handling high-cardinality features, as this is mainly what we presented here - a component which could be added to many architectures, not a full architecture of itself. There is no reason why LMMNN cannot be applied to more complex architectures.
> * Grammar mistakes – if you could point us to 2-3 locations in the text where we have mistakes, that will be appreciated.

---

### Official Review · Reviewer_VuSr · 2021-07-17

**Rating:** 5
**Confidence:** 4

**Summary:**

The paper presented a methodology for dealing with high-cardinality features extracted from categorical variables by using the one-hot encoding applied to deep learning models. The suggested methodology decomposed the model into two sections: fixed effects (FE) and random effects (RE) and it is based on the negative log-likelihood loss from the linear mixed models for neural networks. Presented results including simulated data and four different real data sets with several dozens of categorical variables.

**Main Review:**

-	Originality: The suggested methodology improves other techniques to deal with high-cardinality data such as MeNets and DeepGLMM. The authors established clearly the differences between their approach and benchmarks. The main difference is in the proposed way to estimate the variance of the random effect component. The related work is properly cited by the authors.
-	Quality: Despite authors referenced related work properly, it is not stated why high-dimensional cardinality is a major challenge for Deep Learning models. However, core ideas are properly stated and defined.
-	Clarity: In the suggested methodology is not clear if the inverse of the matrix defined in equation (8) is explicitly computed on their approach. In fact, lines 121-123 state a fast way to compute the inverse of the matrix on small batch sizes, and then lines 127-129 presented an approximation for faster predictions.
-	Significance: Results are important in terms of computing random effects for categorical high-dimensional problems. However, it could be relevant to present other application areas and to illustrate the limitations of deep learning models under high-dimensional cardinality. Besides, the authors present improvement in terms of MSE and standard errors compared to the benchmark, it could be also relevant to indicate the processing time between models to evaluate the level of resources required for their approach.

Questions to be addressed:
- Is the suggested methodology useful for other types of categorical data encoding approaches?
- What is the relationship between high-cardinality and random effects? Is the approach suitable for low-dimensional cardinality?
- How do authors can determine the limit between high and lower dimensional cardinality?
- In the case it is required to compute the inverse of the matrix V, does the suggested approach guarantee the estimated matrix is positive definite?
- In figure 1 it is introduced the notation NLL(f,φ|y) which is not defined in the paper.
- What is the difference between equations (6) and (11)?


**Time Spent Reviewing:**

7 hrs

---

> ### Author Response · Authors · 2021-08-10
> **Comment to Reviewer VuSr**
>
> Thank you for your thoughtful review.
>
> Main review comments:
> * "Quality: why high-dimensional cardinality is a major challenge" – we strongly disagree with this comment, as we believe it is widely understood that high cardinality (informative!) categorical features are a major challenge for learning in general and deep learning in particular, and there is ample evidence in the literature to that extent. We offer some treatment of this subject in Section 4.1 where we discuss one hot encoding and entity embeddings.
> * "Clarity: the V inverse" – the $V$ inverse which appears in (8) is computed “locally” on each batch as is custom in SGD. That is how LMMNN scales LMM. The inverse in lines 127-129 is indeed the inverse of the entire $V$ matrix (or in the case of $N$ over 10K, we used sampling).  This is an additional stage that is necessary for the “prediction” of the $b$, the random effects, as seen in (5). We agree clarity could be improved and will add clarifications if the paper is accepted.
> * "Significance: indicate the processing time between models" – this comment is unclear to us as we **do** show execution time and no. of epochs for all experiments and refer to it (e.g. line 229) in Tables 1-3 in additional materials.
>
> Questions to be addressed:
> * "Is the suggested methodology useful for other types of categorical data encoding approaches?" - This question is unclear to us, but may hold some interesting directions. It would be good to get clarification from the reviewer on the intent here.
> * "Is the approach suitable for low-dimensional cardinality?" - There is no problem in applying the approach for low dimensional cardinality settings, but those are of less interest to us, since they are easily handled by standard approaches. We are specifically interested in the challenge of high-cardinality features.
> * "How do authors can determine the limit between high and lower dimensional cardinality?" - This is related to the previous point: high cardinality is where standard ML approaches get in trouble.  Typically, few hundreds categories, see Section 5.2 with real applications in which $q$ varies from 350 to 40K.
> * "In the case it is required to compute the inverse of the matrix V, does the suggested approach guarantee the estimated matrix is positive definite?" - Yes, as the $V$ discussed in the paper holds a very specific structure (due to $D$’s structure). It is sufficient to have the variance components non-negative (line 111).
> * "In figure 1 it is introduced the notation NLL(f,φ|y) which is not defined in the paper." - The term in figure 1 is $NLL(f, \psi | y)$, we agree it should be $NLL(f, g, \psi | y)$ as in (8)
> * "What is the difference between equations (6) and (11)?" The difference is in the predicted $y_{tr}$ mean of the $j$-th level. In (6), the original LMM, it is the average of the well-known linear predictor $X\hat{\beta_j}$. In (11) it is the average of the (probably non-linear) prediction of the neural net for that level, $\hat{f}(X)$, no betas involved.

---

### Official Review · Reviewer_V1Q6 · 2021-07-28

**Rating:** 4
**Confidence:** 4

**Summary:**

This paper proposes to treat high-cardinality categorical features as the random effects of a mixed non-linear model. The proposed approach, called LMMNN, consists of minimizing the negative log-likelihood over both the functional estimates (parameterized as neural networks) and the variance components of the mixed non-linear model. The framework can be seen as an extension of the random intercepts model. The pipeline is eligible for automatic differentiation, though a closed form can be deduced in simple settings. Experiments are performed on both synthetic and various real datasets.

**Limitations And Societal Impact:**

The limitations and potential negative societal impacts are discussed.

**Main Review:**

The paper is well written with smooth progressions. Related works are well discussed to the best of my knowledge, especially section 4.2 which provides an in-depth comparison with closest previous works. The literature on neural networks over spaces of probability measures (closely related to the estimation of the distribution of random effects) or extensions to non-Gaussian random effects could be mentioned.

The idea to treat categorical features with high cardinality as random effects is pertinent.

However, as is well explained in the paper, the setting is currently limited to continuous explained variables and a single categorical feature to ease computations.

Results seem promising, especially in high-dimensional scenarios. However, LMMNN seems to struggle to estimate variance parameters even in the case of the random intercepts model. Is it partly due to the approximate estimation of V? Are variance estimates more precise as the number of samples increases?

I recommend a reject for this paper, on the grounds that the case study and experiments are limited.

**Time Spent Reviewing:**

2

---

> ### Author Response · Authors · 2021-08-10
> **Comment to Reviewer V1Q6**
>
> Thank you for your thoughtful review.
>
> * "setting is currently limited" – While the current paper does indeed focus on the single categorical variable scenario in the empirical work, it should be clear that the framework is much more general and applies as stated to multiple categorical variables and with minor modifications to kriging/random field scenarios. We will add further clarifications on that if the paper is accepted. The reason we focused on the single categorical setting in this paper is that it is on the one hand conceptually and computationally simple (which we consider an advantage) and on the other hand extremely useful in practice, as our data experiments demonstrate.
> * "LMMNN seems to struggle to estimate variance parameters even in the case of the random intercepts model" – this comment is unclear to us as we have clearly demonstrated in Table 2 very good estimation of the variance components in the random intercepts model, particularly when compared to R’s lmer which is considered SOTA for LMM. The estimation is indeed less accurate in the case of $g(Z) = ZW$ where W is some linear transformation but even then, it is “in the neighborhood” which absolutely cannot be said for lme4.

---

### Official Review · Reviewer_LkaW · 2021-07-28

**Rating:** 4
**Confidence:** 3

**Summary:**

This paper proposes a random effects based objective for deep neural networks. The idea is to modify the standard linear mixed effects objective by replacing the linear predictor with the output of a neural network. The resulting objective can be trained with regular SGD. By modelling a categorical variable (e.g., subject identity) as a random effect, the result is argued to better account for this variable’s influence on the target variable compared to one-hot encoding, and embedding approaches. Experiments on synthetic and real-world datasets demonstrate the feasibility of the proposed approach.

**Limitations And Societal Impact:**

Limitations are adequately discussed.

**Main Review:**

### Overall recommendation

I generally appreciate works that revisit classical statistical tools, particularly those that address pertinent challenges in training modern neural networks. However, I have two main concerns:
- I did not find the proposal insightful – to my understanding, it involves swapping out the linear predictor in a LMM with the output of a neural network. While certainly it is valuable to explicate simple ideas that work well on practically relevant problems, it is hampered by the below point.
- The problem setting considered is fairly restricted: one can employ only a single categorical feature in conjunction with continuous targets and square loss.

Given the above, I am not inclined to recommend this paper for acceptance in its present state.

### Major comments

There are several potentially interesting technical challenges in applying random effects models in a neural context – e.g., moving beyond the square loss, handling multiple categorical variables, modelling hierarchical structure amongst such variables, …. Progress on any of these problems would be welcome. However, the present work is somewhat limited in focus (which the authors are upfront about). In particular, the requirement to use the MSE loss (not the canonical choice in training neural networks for most tasks, barring some recent work [Hui and Belkin, 2020]), and the restriction to using a single categorical feature make the problem setting yet more niche.

The presentation of the mixed effects model is done in generality. This is appreciated, though it seems that all the subsequent applications use the random intercept setting, wherein Z has a very specific structure. It would be useful in this case to explicate, e.g., the closed-form inverse of the matrix V, which I believe exists since the matrix will have a block structure, with each block submatrix being the sum of a (scaled) all-ones matrix plus an identity matrix.

One intuitive property of the mixed effects likelihood (Eqn 3) is that, by virtue of the non-isotropic covariance V, it allows for observations within a group to influence each other’s predictions. Concretely, the objective for β can be seen as adding cross-sample terms with weighting given by V^-1. It would have been nice to see more discussion of this, as the point is somewhat implicit in the present text. This is particularly since one could imagine modifying other losses to encode a similar effect.

One confusion in the framework is that (7) presents g(.) as being an instance of “non-trivial functions which DNNs are suitable for, e.g. non-linear and involving interactions”. However, on L127 and subsequently, it seems that g(.) is the identity function.

In discussing the relation to one-hot encoding, two claims are made:
- It is hard to scale when q is large. What is the precise scalability problem when using SGD based optimisation? Surely even the present method will have a linear scaling with the number of possible categories is large?
- The one-hot features carry little information. What is the additional information that the present method extracts? The paper is not explicit on this point.

One conceptual point that would be worth clarifying is the following. Eqn (2) can be written as y = A w + ε, where A = [X Z] is an n x (p + q) matrix, and w = [ β; b ] is a (p + q) dimensional vector. The matrix A can be seen as adding a one-hot encoding to each input sample’s features. One typically posits a prior distribution N(0, σ2) on β, for which the MAP estimate is implemented by L2 regularisation. One could imagine performing regularised least squares (or indeed, full Bayesian linear regression that marginalises out both β and b) on the matrix A; it is worth commenting on how this differs from the present solution.

In discussing the relation to embedding based methods, it could be useful to comment on modelling relational data (e.g., (user, item) interactions). Here, there is a natural “clustering” and repeated interaction structure in the data (as each user will typically interact with multiple items). The canonical approach to these problems is to apply some form of matrix factorisation, which can be seen as learning user and item embeddings. There is also work on using random effects style models in such settings [Hoff, 2005]. How do these relate to the present work?

It is not clear how precisely the Embedding approach is implemented: in contrast to the one-hot encoding (which is clear), how does the training objective employ the embedding associated with a given category?

When discussing the prior DeepGLMM work, one claim is that the latter is more challenging to train, owing to its use of variational inference. Presumably one could similarly compute MAP/MLE estimates of the underlying model? It was not too clear how the model itself fundamentally differs from the present one.


### Minor comments

- Consider using boldface to denote matrices and vectors – it would make parsing some of the equations easier
- L41: “a categorical feature with possibly thousands of levels — would be considered a random effect variable, often having normal distribution N(0, σ2)” → confusing, not clear how a categorical variable can follow a normal distribution
- L77: cov → \mathrm{cov}
- Eqn 3: consider using \top, not ‘
- L93: _{tr}, _{te} → _{\mathrm{tr}}, _{\mathrm{te}}
- Figure 1: it is not entirely clear what f(X) corresponds to in this figure. Is it the weights on the edges going into the sink node?
- Suppose σb → 0; does the present model reduce to use one-hot encodings?
- Another way of allowing for nonlinearity would be to allow f(X) to follow a Gaussian process. How would this compare to the present approach?
- Would it be interesting to apply a standard LMM to features X that are extracted from a pre-trained neural network? What would be the pros and cons of this strategy?
- Some comments on the relation of the random intercept model to multilevel modelling could be useful. See also [Hoff 2009], [Gelman, 2004; Sec 6].

### References

[Hui and Belkin, 2020] Evaluation of Neural Architectures Trained with Square Loss vs Cross-Entropy in Classification Tasks. arXiV.

[Hoff, 2005] Bilinear Mixed-Effects Models for Dyadic Data. JASA.

[Hoff, 2009] A first course in Bayesian statistical methods. Springer.

[Gelman, 2004] Analysis of variance—why it is more important than ever. The Annals of Statistics.

**Time Spent Reviewing:**

5

---

> ### Author Response · Authors · 2021-08-10
> **Comment to Reviewer LkaW**
>
> Thank you for your thoughtful review.
>
> Overall contribution comments:
>
> * "Swapping out the linear predictor in a LMM with the output of a neural network" is only part of the innovation here. By combining this with the NLL loss, LMMNN could be seen as an alternative way of fitting a non-linear LMM, where the random effects themselves could be learned (with e.g. $g(Z) = ZW$ as demonstrated in simulations). We demonstrate how using SGD and auto-differentiation can scale LMM and reach better predictive performance as compared to existing solutions such as R’s lmer, as well as better variance components estimates. We further provide a Keras implementation one could plug-in to any regression-setting architecture with a high-cardinality categorical feature, and show its performance in various applications.
> * While the current paper does indeed focus on the single categorical variable scenario in the empirical work, it should be clear that the framework is much more general and applies as stated to multiple categorical variables and with minor modifications to kriging/random field scenarios. We will add further clarifications on that if the paper is accepted. The reason we focused on the single categorical setting in this paper is that it is on the one hand conceptually and computationally simple (which we consider an advantage) and on the other hand extremely useful in practice, as our data experiments demonstrate.
>
> Major comments:
> * "Several potentially interesting technical challenges, including moving beyond the square loss, handling multiple categorical variables, modelling hierarchical structure amongst such variable" – As mentioned above, some of these aspects are immediately covered by the current framework (notably dealing with categorical variables and hierarchical structures within the covariance structure). They present some computational challenges but not conceptual ones. On the loss function front, it is important to emphasize that we do not use squared loss, rather we use normal likelihood which is far from squared loss in the LMM case. Of course it can be considered as a generalization of squared loss. Indeed we do plan to move beyond squared loss and regression in future work.
> * "the closed-form inverse of the matrix" – yes, with a single categorical feature with q levels the inverse of $V$ could be seen as $diag(V^{-1}_1, \dots, V^{-1}_q)$ and each of the $V^{-1}_j$ matrices is the inverse of $V_j$ which is $\sigma^2_b \cdot J_n + \sigma^2_e \cdot I_n$ where $J_n$ is an all 1 matrix and $n$ the sample size of this $j$-th level, and $I_n$ the identity matrix. And a similar decomposition could be made for $\log(|V|)$ in the likelihood. That is the reason why SGD could be proven to work here if we’re willing to assume each batch is roughly the sample size of the $j$-th level. We agree this slightly more rigorous part is missing and will add it to the paper if it is accepted. intend to add it in future work.
> * " $\beta$ can be seen as adding cross-sample terms with weighting given by $V^{-1}$" – The “cross terms” play two different roles – one in estimating $\beta$ (or here the network params), where it is relatively minor, and one in the BLUP estimation, where it is very major as can be seen from the BLUP formulas and the resulting b estimates in Table 1 in additional materials
> * "on L127 and subsequently g(.) is the identity function" – this is not accurate, we show a simulation (Tables 2,3) where $g$ is an “embedding” of $Z$, in which LMMNN performs best by far.
> * OHE claims:
>     * "hard to scale when q is large" – the meaning is that the naïve approach of building an indicator matrix with $q$ columns where $q$ is over 10K and $N$ is in the area which is of interest to us (say 100K and over) is simply not feasible without some federated way of storing and training the data. Furthermore, overfitting is a danger in this scenario where we fit a weight for each of these q levels. In contrast LMMNN does scale with $q$s even over 100K because of smart sparse implementation and assuming only a single parameter $sigma^2_b$ controlling the normal distribution of the $q$ REs
>     * "one-hot features carry little information" – e.g. the distance or relation between two OHE features could either be “same” or “different”, as opposed to entity embeddings or using LMMNN with $g(Z) = ZW$ where $W$ is some linear transformation as demonstrated which, like embeddings, maps the 0/1 vectors into a lower space where they could also be “close” or “distant”
> * "Bayesian linear regression and regularization" – we are aware that even not in the context of neural networks one can view random intercepts for instance as a special case of ridge regression. The LMMNN view offers two important distinctions from the ridge regression or Bayesian view: first, it separates the variables into the two families (fixed and random effects) which are treated differently; and more importantly, the LMM likelihood framework allows treating the regularization parameter as part of the optimized parameters (called variance components in the LMM context), and hence fit the best regularization on the training data. As our results show, this works extremely well in both simulations and real data.  Once we go beyond the random intercept model (into the $g(Z) = ZW$ embedding or to more complex models) the ridge analogy breaks down, but the LMMNN approach remains valid.
> * "modelling relational data" – we have read the Hoff 2005 which deals with dyadic data such as seen in social networks and naturally applies LMM and mixed-effects ANOVA-like decompositions to handle these including modeling interaction effects between pairs of units. For some applications we have discussed, mainly repeated measures on the same subject, data could be modeled this way, for example for the repeated facial images of people in the CelebA dataset. For other applications, our clustering variable is just another feature, like the host of an Airbnb apartment or the job of a subject in the UK Biobank dataset. Furthermore, our ultimate goal is to handle LMM in general in deep neural networks, not just the case of a single clustering unit. Finally, we are interested in high dimension datasets, where $N$ is large and $q$, the number of levels in the clustering variable, can reach over 100K, which would be less suitable for the types of scenarios addressed in Hoff 2005.
> * "not clear how precisely the Embedding approach is implemented" – we feel this is standard and did explain in section 4.1 “feeding a neural network with feature $v$ after it has been one-hot encoded into a $q$-length vector, and using the network's inherent loss function and back propagation algorithm to learn a low-dimensional representation of length $d$ (typically $d << q$)”. So embeddings here are a $q \times d$ matrix from $Z$ to $d$-dimensional continuous vectors.
> * DeepGLMM – As we noted in section 4.2 we do acknowledge DeepGLMM’s model to be similar to ours, there are two critical differences: 1.  DeepGLMM is applicable only to “random slopes” or temporal/longitudinal scenarios whereas we aim to handle mixed effects models in general, including that of temporal dependence which we are now exploring. 2. DeepGLMM’s computational approach and implementation involves a few layers of complexity which makes it less accessible to researchers and unseemly to apply to big data . Our implementation could be plugged-in to many models with 1 or 2 additional lines of code and deals with modern data sizes with no difficulty.

---

> > ### Comment · Area_Chair_8WeP · 2021-08-10
> > **from the area chair**
> >
> > Dear authors: As the area chair, I appreciate this paper and its contribution. Therefore, I would like to suggest how to strengthen your response. It says in several places "We will add further clarifications ... if the paper is accepted." Please add those explanations now, in this conversation. Topics to cover include:
> > - multiple categorical variables
> > - "if we’re willing to assume each batch is roughly the sample size of the _j_-th level"
> > - "... relatively minor ... very major ..."
> > - "smart sparse implementation"

---

> > > ### Author Response · Authors · 2021-08-10
> > > **Clarifications to authors comment**
> > >
> > > * multiple categorical variables: we claim our approach can be easily generalized to the case of multiple categorical variables, and in fact have done so in the period since the original paper submission with good results, only on simulated data, only with $g$ as identity function for now. We could add those results to the original paper given more space or in additional materials. Simulated data meaning say two categorical features, one with say $q_1 = 10000$ levels and one with $q_2 = 1000$ levels, one producing $b_1$ random effects with say $\sigma^2_{b1}=0.1$ and one producing $b_2$ random effects with $\sigma^2_{b2} = 10$. The main current challenge we are facing is a theoretical one: as opposed to the situation with a single categorical variable where we can roughly establish why using SGD on the NLL loss is valid (see next point) and the estimated variance components converge to the optimum, here we need further research to understand why this is working. The covariance $V$ matrix is no longer block diagonal (see next point) but the **sum** of block diagonal matrices, e.g. in the above case of two categorical features: $V = \sigma^2_{b1}Z_1Z_1' + \sigma^2_{b2}Z_2Z_2' + \sigma^2_{\varepsilon}I$. But as said we get good results and need further research.
> > > * "willing to assume each batch is roughly the sample size of the $j$-th level": with a single categorical feature with $q$ levels let $\theta$ be the variance components $[\sigma^2_b, \sigma^2_e]$ and we can write: $V(\theta) = \sigma^2_b ZZ' + \sigma^2_{\varepsilon} I_n$. This means $V$ is block diagonal if the observations are ordered by that single categorical feature: $V(\theta) = diag(V_1, \dots, V_q)$, where each block $V_j$ is of size $n_j \times n_j$ and $V_j(\theta) = \sigma^2_b J_{n_j} + \sigma^2_{\varepsilon} I_{n_j}$ where $j = 1, \dots q$ and $n_j$ is the number of observations for RE level $j$. This means we can write the inverse as block diagonal as well: $V(\theta)^{-1} = diag(V_1^{-1}, \dots, V_q^{-1})$. And we can re-write the log-determinant as a sum: $\log{|V(\theta)|} = \log(|V_1| \times, \dots, \times |V_q|) = \sum_{1}^{q}\log|V_j|$. Furthermore the NLL in (8) can be decomposed to a sum over $1,\dots, q$:
> > > $NLL(f, \theta | y) = \sum_{j = 1}^{q}\{[\frac{1}{2}(y_j - f(X_j))'V_j(\theta)^{-1}(y_j - f(X_j)) + \frac{1}{2}\log{|V_j(\theta)|} + \frac{n_j}{2}\log{2\pi}]\}$
> > >
> > >     And the gradient could be written as the sum of gradients:
> > > $\frac{\partial NLL}{\partial \theta} = \sum_{j = 1}^{q}\{[-\frac{1}{2}(y_j - f(X_j))'V_j(\theta)^{-1}\frac{\partial V_j(\theta)}{\partial \theta}V_j(\theta)^{-1}(y_j - f(X_j)) + \frac{1}{2}tr(V_j(\theta)^{-1}\{\partial V_j(\theta)}{\partial \theta})]\}$
> > >
> > >     Therefore if each level had the same $n_j = m$ and each batch of size $m$ would have been taken only from the $j$th level, we have essentially proven why SGD works. In real applications this is of course not the case, and sometimes multiple groups enter a single batch. But we do sort the data according to the single categorical feature as an approximation which shows good convergence behavior.
> > >
> > > * "... relatively minor ... very major ...": In the LMM estimator for $\beta$, $(X'\hat{V}^{-1}X)^{-1}X'\hat{V}^{-1}y$, we can see how $V$ is essentially weighting the OLS well-known estimator, like in generalized least squares. This is what we mean by "relatively minor role", and here of course we have no $\beta$, but the neural networks weights, and this nature of weighing is yet unclear. In the BLUP for the RE $b$, the $V$ matrix has a "major role" as can be seen in (5) or its adaptation in LMMNN: $\hat{b} = D(\hat{\theta})\hat{g}(Z_{tr})'V(\hat{\theta})^{-1}(y_{tr} - \hat{f}(X_{tr}))$, without it we cannot predict $b$ and improve prediction for seen groups/levels. We further specified the quality of this prediction of $b$ can be seen in both Table 2 in the paper and the predicted vs. true scatterplots in additional material Figure 1.
> > >
> > > * "smart sparse implementation": The way one goes about one-hot encoding is to map each level of a $q$-levels categorical feature into its own 0/1 feature or column. This quickly does not scale when $q, N$ are very large. In our Keras implementation for LMMNN (in additional material and will be available in Github) there is no need to create the $Z$ one-hot matrix like that, the usual **single** column (with  $q$ levels typically mapped to $0, \dots, q-1$) is mapped in each batch to a sparse Z matrix using tensorflow `tf.sparse.SparseTensor` class making LMMNN relatively fast (see additional materials) and scale with $q$ way over 10K.
> > >
> > > We are happy to provide any more clarifications.

---

> > ### Comment · Reviewer_LkaW · 2021-08-25
> > **Follow up comments**
> >
> > I'd like to thank the authors for their detailed reply.
> >
> > *Core innovation*
> >
> > I gather the authors' point is that a classic LMM has y = X w + Z b + e, while the present work has y = f(X; w) + g(Z; b) + e, where f, g can be neural networks. My original statement focussed on the former, while the response indicates the latter is also important. I agree this was imprecise, but the core concern about modest insight stands: each of the original linear terms in the LMM is swapped out with a neural network, in a somewhat restricted setting.
> >
> >
> > *Multiple categorical variables*
> >
> > My comment about the restriction to a single categorical variable followed the para starting L317 of the paper. This acknowledges a limitation of the paper (which is appreciated), namely, the focus on the single categorical variable case. The para also hints at the possibility of extending it while noting that "more theoretical work is needed here". The response provides further context on the resultant challenges, which is appreciated. However, it seems that the core issue remains that the extension possesses theoretical challenges, and its performance on real-world problems is unclear.
> >
> >
> > *Square loss*
> >
> > I acknowledge that my original statement was imprecise, but the core concern stands. A more precise statement, which gets at the core of the concern, is that the focus of the paper is on the regression as opposed to classification setting (i.e., y is assumed to be a real number with a suitable Gaussian distribution, as opposed to a discrete value with a suitable categorical distribution parameterised by a softmax).
> >
> >
> > *Scaling with q is large*
> >
> > I do not follow why one cannot use similar tricks to train a OHE model when q is large. Doesn't the case σb → 0 correspond to using OHE?
> >
> >
> > *Embeddings*
> >
> > I did read the discussion in Section 4.1, but unfortunately couldn't translate it to an equation. I gather that OHE would involve positing y = X w + Z b + e, where b is treated as a fixed vector of length q. Would the embedding approach posit y = X w + Z E b' + e, where E is a learned matrix of size q x d, and b' is a fixed vector of length d?
> >
> >
> > *Comparison to DeepGLMM*
> >
> > Regarding computational complexity, is the core issue that DeepGLMM uses full Bayesian inference, while the present paper is more akin to type-II maximum likelihood or empirical Bayesian inference? My original question aimed to understand whether applying type-II MLE to DeepGLMM would yield a similar objective to the present paper. Would the main difference be that its objective is restricted to the random slopes setting?

---

> > > ### Author Response · Authors · 2021-08-27
> > > **Comment 2 to Reviewer LkaW**
> > >
> > > We appreciate the reviewer’s thoughtfulness and effort and the chance to have ongoing interaction (regardless of whether it leads to an upgrade of our paper’s scores). We do feel however that some of the reviewer’s points still do not do our paper and its contribution justice.
> > >
> > > *Core innovation*
> > >
> > > This is the main point. The reviewers’ reference “in a somewhat restricted setting” is unclear and does not seem justified. The function $f(X)$ is a general function of $X$, non-linear and non-additive, and our LMMNN approach successfully fits such functions with complex interactions, as demonstrated in the simulations. Perhaps the reviewer is concerned about the additive separation between $f$ and $g$? This may appear somewhat limiting, but is not really so, as $X$ and $Z$ can overlap, and $f$ and $g$ can each be rather complex in principle, or they can even be identical as in MeNets. None of these combinations is problematic for our implementation approach. The ability of LMMNN to model well complex $f$ and $g$ also depends on the neural network architecture(s) used, here we have demonstrated the use of a standard multi-layer perceptron, a convolutional neural network and LSTM.
> > >
> > > *Multiple categorical variables*
> > >
> > > We acknowledge that there are remaining implementation challenges when moving to multiple categorical variables, but the foundation is there, specifically there are no theoretical challenges. This is detailed in our previous response and can be incorporated to the paper, replacing or augmenting that comment in L317. From a practical perspective we feel it goes without saying that if the single categorical LMMNN version is useful, the multiple categorical version would be even more useful.
> > >
> > > *Square loss*
> > >
> > > We agree, this is a “regression first” paper. We would like to lay the foundation to more settings here but we can make it even clearer in the paper.
> > >
> > > *Scaling when q is large*
> > >
> > > The scaling of linear one-hot encoding could be done in a similar way, non-linear OHE would be more challenging. Notwithstanding, we wanted to address the standard way practitioners use to implement OHE, which, as $q$ goes large, will consume more memory and will not scale, whereas our implementation assumes $q$ can be very large in the first place and seamlessly allows it. Finally, even if implemented in a way which allows it to scale, OHE would still mean learning a (differentiable) weight for each and every level of $q$, which is prone to overfitting.
> > >
> > > *Embeddings*
> > >
> > > Yes, the usual way to think of embeddings is multiplying a $Z_{n \times q}$ with a (learned) $E_{q \times d}$ matrix where $d << q$, then proceeding with this $d$-length input to the neural network. We can rephrase this but again feel this is standard practice when it comes to embeddings, it is not in the scope of this paper.
> > >
> > > *DeepGLMM*
> > >
> > > There are two main differences between LMMNN and DeepGLMM, one theoretical, one concerning implementation. The theoretical one, as the reviewer writes and as we write in Section 4.2 – DeepGLMM’s goal objective is restricted to random slopes.  For LMMNN, this is one of the “extensions” we’re looking into, and believe that applying the negative maximum likelihood loss in the simple unified way we show to this objective should reach similar performance. The second difference which we insist is substantial is regarding implementation. Our work is a plug-in solution, written concisely and can be used with little to no accommodation e.g., in a model built with standard Keras, as we demonstrate in additional materials. DeepGLMM’s many layers including importance sampling, factor covariance and variable selection, are far from “easy to use” and any change in architecture would require the practitioner to dig deeply into the training of the network.

---

> > > > ### Comment · Reviewer_LkaW · 2021-08-27
> > > > **Re: Comment 2 to Reviewer LkaW**
> > > >
> > > > Thanks for the detailed reply.
> > > >
> > > > I wanted to clarify the concern regarding "in a somewhat restricted setting". This was not referring to the decomposition of y as f(X) + g(Z) being somehow limited in capacity. The comment instead was referring to the restriction to regression settings with a single categorical variable. I apologise for any confusion.
> > > >
> > > > Regarding OHE, I certainly agree that from a modelling perspective, it is less powerful than a random intercept model. My confusion on this issue -- which was minor, and did not substantively affect my score, but I think would nonetheless be good to clarify in an updated version -- was whether there is something _inherent_ in such a model that prevents scaling to the large q setting. If I've understood the response, the claim is that one could make OHE scale using similar tricks as done for the present method; but this would deviate significantly from the common implementation, which does not scale. If this is indeed so, I would suggest to slightly reword the text to make this point clearer.
> > > >
> > > > To give further context: my confusion also stemmed from the fact that in settings such as recommender systems, it is common to have user- and item-specific bias terms, which can be seen as using OHE. The # of users and items can easily be in the 100K+ range, and so I was not sure as to the claim of scalability.

---

### Decision · Program_Chairs · 2021-09-27

**Decision:**

Accept (Poster)

**Comment:**

As the area chair, I recommend that this paper be accepted and published. The main reason is that the paper is genuinely useful to practitioners, because it addresses an issue that arises in many applications and the solution is clear and simple.

A paper like this one should not be rejected because the authors have failed to do some work that reviewers would like to see. That work should be in future papers. The authors are clearly knowledgeable and competent. The technical work is of good quality and correct. Practitioners and future researchers can use and build on this.

In the final version, hopefully the authors will address the last part of the following comment from an expert reviewer: "High-cardinality categorical variables are pervasive in practice. An effective, generic, clearly expressed solution to coping with such variables would certainly be a welcome contribution to the literature. This is true even if the solution has modest technical novelty; indeed, it would be a virtue if one could arrive at a simple solution based on existing ideas. One could make a case that the authors' solution is effective. However, I am less sure about generic and clearly expressed..."